# Targeting Atherosclerosis via NEDD4L Signaling—A Review of the Current Literature

**DOI:** 10.3390/biology14030220

**Published:** 2025-02-20

**Authors:** Lucas Fornari Laurindo, Victória Dogani Rodrigues, Enzo Pereira de Lima, Beatriz Leme Boaro, Julia Maria Mendes Peloi, Raquel Cristina Ferraroni Sanches, Cláudia Rucco Penteado Detregiachi, Ricardo José Tofano, Maria Angelica Miglino, Katia Portero Sloan, Lance Alan Sloan, Sandra Maria Barbalho

**Affiliations:** 1Department of Biochemistry and Pharmacology, School of Medicine, Universidade de Marília (UNIMAR), Marília 17525-902, SP, Brazil; 2Postgraduate Program in Structural and Functional Interactions in Rehabilitation, School of Medicine, Universidade de Marília (UNIMAR), Marília 17525-902, SP, Brazil; 3Department of Biochemistry and Pharmacology, School of Medicine, Faculdade de Medicina de Marília (FAMEMA), Marília 17519-030, SP, Brazil; 4Postgraduate Program in Animal Health, Production and Environment, School of Veterinary Medicine, Universidade de Marília (UNIMAR), Marilia 17525-902, SP, Brazil; 5Department of Animal Anatomy, School of Veterinary Medicine, Universidade de Marília (UNIMAR), Marília 17525-902, SP, Brazil; 6Texas Institute for Kidney and Endocrine Disorders, Lufkin, TX 75904, USA; 7Department of Biochemistry and Nutrition, School of Food and Technology of Marília (FATEC), Marília 17500-000, SP, Brazil

**Keywords:** atherosclerosis, NEDD4L, blood vessels, inflammation, oxidative stress, cardiovascular diseases, cardiovascular outcomes, atherosclerotic cardiovascular diseases

## Abstract

Atherosclerosis remains in the arteries. This disease is characterized by the buildup of fat plaques within those vessels, causing them to narrow, which ultimately obstructs their interior. Atherosclerotic cardiovascular outcomes, including stroke and ischemic heart disease, are leading causes of death worldwide, and the incidence increases as the population grows and ages. NEDD4L is significantly associated with the occurrence of atherosclerosis. However, NEDD4L’s effects on atherosclerosis plaque formation are dubious, and no previous review has summarized its actions. Therefore, we aimed to fill this gap. Our results mainly demonstrate that NEDD4L is detrimental, increasing atherosclerosis occurrence. However, other studies indicated that stimulating NEDD4L may positively counter atherosclerosis plaque formation. Therefore, future research endeavors must address several limitations. More research based on preclinical studies must be conducted to translate findings into clinical trials successfully.

## 1. Introduction

Cardiovascular diseases are the leading cause of morbidity and mortality worldwide. As the population grows and ages, ischemic heart disease, stroke, and many other diseases are increasing in frequency, which ultimately leads to an increase in disability-adjusted life years, which represents the number of years a person could have lived without disabilities caused by preventable events, such as cardiovascular events [1,2]. Although many risk factors for cardiovascular diseases have generally been declining among populations, including smoking, hypertension, and dyslipidemias, acute treatment strategies have been less well-funded, and secondary prevention has been extensively delivered. Other factors intimately associated with cardiovascular disease burden, like obesity and diabetes, have increased among countries, which correlates with atherosclerosis, a significant contributor to cardiovascular physiopathology, having a detrimental effect on the trends in cardiovascular disease prevention and intervention [3].

Atherosclerosis is a chronic inflammatory disease that affects arteries and is the cause of approximately 50% of all deaths in Westernized countries. This chronic condition is characterized by low-density lipoprotein (LDL) and remnant lipoprotein particle deposition, leading to disturbed non-laminar flow at branch points in the arteries, which is the primary cause of atherosclerotic cardiovascular disease such as the aforementioned ischemic conditions [4]. The risk of atherosclerotic cardiovascular disease is highly heterogeneous in both healthy and unhealthy populations, and it is a convention that preventive cardiology begins with assessing baseline atherosclerotic cardiovascular disease risk. Additionally, lipid-lowering treatment strategies are often insufficient [5,6]. However, some genes are indissociably associated with atherosclerosis physiopathology, and understanding their effects on the disease’s occurrence could undoubtedly define effective screening and treatment strategies [7]. One such gene is the neural precursor cell-expressed developmentally downregulated 4-like (NEDD4L) gene [8].

NEDD4L is related to ubiquitin ligase (E3) enzyme activities and is essential in human physiology and pathology. NEDD4L often carries out K63-mediated ubiquitination of their targets, primarily through signaling and/or the endocytosis/trafficking of transmembrane proteins associated with lysosomal degradation. In particular, NEDD4L regulates cellular growth and proliferation and acts during cardiovascular diseases and outcomes, including hypertension, cardiac remodeling, and atherosclerosis [9]. Emerging evidence suggests ubiquitination is an essential factor in atherosclerosis pathogenesis in different capacities, including regulating vascular inflammation, atherosclerosis plaque stability, endothelial and vascular smooth cell function, and lipid metabolism [10]. Nevertheless, despite previous efforts, no review has been published targeting the role of the NEDD4L gene in atherosclerosis plaque formation. Herrmann et al. [11] reviewed the roles of protein metabolism and degradation as a central element of atheroma progression. However, they did not intervene with NEDD4L ligands or mention this target gene in their paper. At the same time, Wilck et al. [12] also reviewed the activities of the ubiquitin–proteasome system against atherosclerosis, but their analysis also lacked intervention with NEDD4L. In addition, Herrmann et al. and Wilck et al. published their manuscripts during the initial part of the last decade, limiting their applicability in the light of recent findings. Recently, Poels et al. [13] mentioned NEDD4L in their paper, highlighting the role of gene interactions in limiting cholesterol efflux from macrophages but limiting the investigation to only one aspect of NEDD4L activities against atherosclerosis occurrence and progression.

Given the limitations of previous studies and the call for further research, this review takes a proactive stance. It systematically explores the impact of NEDD4L signaling on atherosclerosis occurrence and progression, integrating preclinical in vitro and in vivo models due to the current absence of clinical trials. This review systematically examines relevant studies by providing a comprehensive overview of NEDD4L’s potential as a therapeutic target. More importantly, it critically addresses the limitations of current research and suggests future directions for advancing NEDD4L-based therapies against atherosclerosis. This detailed exploration aims to enhance the understanding of NEDD4L’s multifaceted role in modulating atherosclerosis-related pathways. Although NEDD4L is closely related to cardiovascular disease occurrence, evidence underscoring the relationship between NEDD4L signaling and atherosclerosis is primarily new. Additionally, until 2024, when three contemporary studies in the field were published, the evidence of the role of NEDD4L in atherosclerosis lacked comprehensive detail. Therefore, considering these present-day discoveries and the research gaps mentioned above, the importance of this emerging field of study has been rekindled, and a systematic review addressing this topic has become necessary. Finally, this review synthesizes existing knowledge and provides a foundation for future research to translate these findings into clinical applications. Figure 1 depicts our study’s rationale.

## 2. An Overview of Atherosclerosis Physiopathology with an Emphasis on NEDD4L Signaling

Atherosclerosis is a chronic inflammatory disease that affects blood vessels and is the primary cause of cardiovascular diseases worldwide. Recently, computational bioinformatics analysis identified NEDD4L as a key target gene associated with atherosclerosis occurrence, progression, and treatment [14]. Our work may provide a scientific background for the pathogenesis and treatment of atherosclerosis based on NEDD4L targeting.

### 2.1. Atherosclerosis: A Chronic Inflammatory Condition Predisposing to Cardiovascular Diseases and Outcomes

Atherosclerosis is a chronic inflammatory condition and the principal pathological cause of atherosclerotic cardiovascular diseases. It is characterized by the thickening of the arteries, caused by the formation of plaques composed of fatty acids, cholesterol, calcium, fibrin, cellular debris, and other compounds, particularly in regions of non-laminar flow, that is, at the branching points of the arteries. Endothelial dysfunction in areas of the arterial vasculature prone to injury can profoundly contribute to the pathobiology of atherosclerotic cardiovascular disease [15,16,17]. Different degrees of arterial stenosis can obstruct blood flow, causing hypoxia in vital organs such as the heart, brain, kidneys, pelvis, arms, and lower extremities [18,19].

Risk factors include hypercholesterolemia, hypertension, diabetes mellitus, smoking, age, gender, family history, sedentary lifestyle, obesity, diets rich in saturated and trans fatty acids, and specific genetic mutations [20,21,22,23,24,25].

The pathophysiology of atherosclerosis is mainly due to the continuous process of damage to the arterial wall due to lipid retention, which causes chronic inflammation in vulnerable areas of the arteries. This lipid trapping creates fatty streaks in the arterial intima, which subsequently develop into fibrous plaques and complex atherosclerotic lesions, which are susceptible to rupture. In addition, arterial stenosis caused by the shortening of the lumen by atherosclerosis plaques can result in the occlusion of vessels, such as the coronary arteries [25,26,27]. More specifically, areas prone to the development of atherosclerosis show upregulation of nuclear factor kappa B (NF-κB), a pivotal player in the inflammatory cascade. This occurs because LDL particles leave the blood, enter the arterial intima, accumulate, and undergo oxidative modification, creating oxidized low-density-lipoprotein (oxLDL) taken up by scavenger receptors (SRs) and forming foam cells. In this sense, along with the endothelial dysfunction caused by the risk factors, there is also a deficiency in nitric oxide (NO) and prostacyclin (both vasodilators) and/or an increase in cell adhesion molecules (CAMs), causing the accumulation of monocytes and lymphocytes. The monocytes, in turn, mature into resident macrophages through macrophage colony-stimulating factor (M-CSF), taking up the oxLDL and intensifying the formation of foam cells [28,29]. Moreover, many other mechanisms have been implicated in atherosclerosis, such as chronic endoplasmic reticulum stress and mitochondrial dysfunction [29,30,31,32].

Macrophage polarization also plays a vital role in the formation of atherosclerotic lesions. For example, M1 macrophages are implicated in the pro-inflammatory effect, secreting interleukin one beta (IL-1β), interleukin 6 (IL-6), and tumor necrosis factor-alpha (TNF-α). In contrast, M2 macrophages secrete interleukin 10 (IL-10) and transforming growth factor beta (TGF-β) as anti-inflammatory cells [33,34,35,36,37]. As pointed out, NEDD4L promotes the polarization of M1 macrophages, which are closely linked to inflammatory processes, especially in the vascular endothelium. It also increases the uptake of oxLDL and the formation of foam cells by the ubiquitination of the nuclear factor of kappa light polypeptide gene enhancer in B-cells inhibitor alpha (IκBα)/peroxisome proliferator-activated receptor gamma (PPARγ) and the phosphorylation of the suppressor of mothers against decapentaplegic (SMAD) family member 1/SMAD family member 2 (SMAD1/SMAD2), accelerating the progression of atherosclerosis [38,39].

In this sense, foam cells promote the accumulation of leukocytes and the secretion of cytokines and growth factors, which accelerate the inflammatory response in addition to also promoting the accumulation of smooth muscle cells under the stimulus of angiotensin II, platelet-derived growth factor (PDGF), insulin-like growth factor (IGF), and others, forming fibrous plaque enriched with collagen [40,41,42]. However, with the action of proteolytic enzymes such as matrix metalloproteinases (MMPs), this fibrous plaque is degraded, favoring the formation of a thrombus through the activation of tissue factor and platelet aggregation that extends to the arterial lumen, increasing the risk of rupture and life-threatening thrombosis [4,43,44].

### 2.2. General Aspects of the NEDD4 Signaling Pathway

NEDD4L is a member of the neural precursor cell-expressed developmentally downregulated 4 (NEDD4) family of E3 enzymes. This family is one of the main groups of E3 enzymes, including NEDD4, NEDD4L, ITCH, SMURF1, SMURF2, WWP1, WWP2, NEDL1, and NEDL2 [45]. This group of homologous to E6-AP C-terminus (HECT) E3 enzymes was initially cloned in the mouse central nervous system, containing an N-terminal C2 domain (calcium-dependent lipid binding domain), three (rat or mouse rat) or four (humans) WW domains (interaction domains of protein–protein), and the HECT domain. NEDD4 or NEDD4-1 was the first to be discovered and has been extensively investigated [46]. Some potential NEDD4 substrates, such as the sorting adaptor Hgs, have been described, suggesting that they may have a role in vesicular sorting and trafficking [46,47,48,49,50].

NEDD4L regulates several proteins that regulate the epithelial Na^+^ channel, helping maintain sodium balance, deoxyribonucleic acid (DNA) repair, autophagy, and antiviral immunity and ensuring cellular homeostasis [51,52,53]. NEDD4L is known to promote angiogenesis. Abnormalities in this process can lead to several diseases, including cancer [54,55], diabetic retinopathy [56], and vascular malformations [38,57]. In addition, this ligase also acts in some human diseases, such as diabetic kidney disease (in diabetic nephropathy, NEDD4L increased high-glucose (HG)-induced podocyte inflammatory injury) [58], diabetes, and atherosclerosis [59,60,61,62].

Some authors have shown that, in vivo, NEDD4 proteins possess multiple targets. Abnormalities or deficiency of some of these molecules may result in growth retardation, nervous and cardiovascular system developmental abnormalities, and the impairment of neuromuscular junctions and T-cells (NEDD4 regulates their function) [63,64]. Furthermore, animals with NEDD4 knockout show growth retardation (due to imbalanced cell surface expression and signaling mediated by insulin and IGF receptors), and NEDD4 knockout interferes with neurite growth and neuron arborization [65,66]. Moreover, NEDD4 is essential for vascular development [67], being responsible for ubiquitinating and degrading N- and c-Myc oncoproteins in neuroblastoma and pancreatic cancer cells [54,55,56], and can interact with viral proteins to mediate the budding of many viruses [68,69].

### 2.3. Exploring Atherosclerosis Physiopathology with an Emphasis on NEDD4L Signaling

Some studies have proposed an association between NEDD4L, atherosclerosis, and other cardiovascular diseases. It has been discovered that NEDD4L overexpression can accelerate the migration, angiogenesis, and proliferation of human umbilical vein endothelial cells (HUVECs), facilitating the development of atherosclerosis [53,70]. Also, in oxLDL-induced HUVECs, Xu et al. [71] demonstrated that the expression of NEDD4 is rapidly reduced. After NEDD4 overexpression, the inflammatory process, endothelial cell dysfunction, and damage were attenuated. NEDD4 and NEDD4L are both ligases. However, they have opposite effects on endothelial cells. While the first reduces atherosclerosis development through reduced inflammation and endothelial cell dysfunction, the latter influences endothelial cell migration, angiogenesis, and proliferation and may contribute to atherosclerosis development.

Song et al. [33] investigated the role of micro ribonucleic acid (miR)-30a-5p in both in vivo and in vitro atherosclerosis. They highlighted that the anti-atherosclerotic actions of miR-30a-5p were modulated by targeting NEDD4L. Additionally, they demonstrated that the overexpression of miR-30a-5p promoted a significant reduction in atherosclerosis by reducing pro-inflammatory biomarkers and increasing the anti-inflammatory response, regulating lipid uptake and the M1/M2 macrophage phenotype. Therefore, NEDD4L participated in regulating lipid metabolism and in the transition of the M1/M2 macrophage phenotype.

Another interesting study investigated homocysteine’s role in regulating the proliferation, migration, and phenotypic modifications of vascular smooth muscle cells (VSMCs) through sirtuin-1 (SIRT1)/signal transducer and activator of transcription 3 (STAT3) via the NEDD4L E3 ubiquitin-protein ligase WWP2. Their results demonstrated that WWP2 induced the proliferation, migration, and phenotype switching of homocysteine-induced VSMCs through upregulating SIRT1/STAT3 signaling phosphorylation [72].

Hu et al. [73] showed that the overexpression of human lipocalin-2 in hepatocytes reduces atherosclerosis development via scavenger receptor group B type 1 (SR-B1) in western-diet-fed LDL receptor (LDLR) ^−/−^ mice. The hepatocyte-specific ablation of lipocalin-2 promoted the opposite effect. Hepatocyte lipocalin-2 improves high-density lipoprotein (HDL) metabolism and reduces atherogenesis due to the inhibition of NEDD4-1-mediated SR-B1 ubiquitination.

Chen et al. [61] showed that Ras guanyl-releasing protein 2 (RASGRP2) overexpression inhibited the permeability of HG- and oxLDL-induced human cardiac microvascular endothelial cells (HCMECs). It also inhibited the formation of reactive oxygen species (ROS) and apoptosis. On the other hand, it speeds up cell viability, migration, and angiogenesis. Furthermore, NEDD4L can interact with RASGRP2 through ubiquitination, leading to the inhibition of RASGRP2 stability. Adipose-derived mesenchymal stem cells (ADMSCs) exosomes overexpressed RASGRP2 and induced cell migration, viability, and angiogenesis, reducing permeability, apoptosis, and ROS production in the induced HCMECs. Due to these results, the authors conclude that targeting the NEDD4L/RASGRP2 axis or inducing RASGRP2-modified ADMSCs exosomes could be a strategy for alleviating diabetes-related atherosclerosis.

In the study by Chen et al. [38], the results showed that exosomal NEDD4L knockdown could inhibit M1 polarization, oxLDL uptake, and the formation of foam cells due to a reduction in the protein levels of p-IκBα/IκBα, p-SMAD1/SMAD1, p-SMAD2/SMAD2, and p-P65/P65.

Furthermore, NEDD4L also acts in the modulation of NO. NO deficiency is linked to endothelial dysfunction and is associated with hypertension and atherosclerosis pathophysiological processes. However, it is known that the overexpression of NEDD4L-promoted protein kinase b (Akt) favors the phosphorylation of endothelial nitric oxide synthase (eNOS), which can impair the action of this enzyme, ultimately favoring the formation of atherosclerosis plaques under hyperlipidemic conditions [53,74,75]. NEDD4L is also associated with hypertension and miR-30a-attenuated atherosclerosis [33,76,77].

Some authors investigated the role of NEDD4L in regulating endothelial cell angiogenesis and the possible mechanisms involved. They observed that the gain and loss of function of NEDD4L predicted angiogenesis and played a role in the mobility of HUVECs. NEDD4L knockdown inhibited tube formation, cell proliferation, and the migration of HUVECs. Notwithstanding this, NEDD4L could regulate the angiogenesis and cell progression related to the phosphorylation of Akt, extracellular signal-regulated kinase (ERK) 1/2, and eNOS, and vascular endothelial growth factor receptor 2 (VEGFR2) and cyclin D1 and D3 expression (the NEDD4L/Akt/ERK/Cyclin D1/3 axis is critical in angiogenesis regulation). This research showed evidence that NEDD4L is essential for promoting angiogenesis in HUVECs. In addition, pharmacological or genetic promotion of NEDD4L may represent a promising therapeutic strategy for treating ischemic diseases, such as myocardial infarction, which exacerbates diabetic retinopathy and critical limb ischemia, by promoting an angiogenic role in the ischemic tissues [53].

Based on the above, we suggest that NEDD4L has a multifunctional role in the body, playing a critical role in biochemical, physiological, and pathological pathways. Pharmacological or genetic NEDD4L modulation may be a promising therapeutic strategy that can be considered in the treatment of ischemic diseases, such as myocardial infarction, exacerbation of diabetic retinopathy, and limb ischemia, due to the stimulation of angiogenesis in the ischemic tissues [53]. Figure 2 elucidates atherosclerosis physiopathology, emphasizing the role of NEDD4L signaling.

## 3. Implications of Targeting NEDD4L Signaling Against Atherosclerosis

Table 1 illustrates the multifaceted roles of NEDD4L in atherosclerosis occurrence and progression. Although the evidence is heterogeneous, most studies demonstrated that NEDD4L has detrimental effects on atherosclerosis plaque formation. This section also depicts the literature search methodology and reporting method to ensure rigor and consistency in communicating our findings.

### 3.1. Literature Search Methodology

We employed a systematic literature search methodology to comprehensively review the impact of NEDD4L signaling on atherosclerosis physiopathology across different models. We focused solely on preclinical studies due to the absence of clinical trials in this domain. Our main objective was to identify, evaluate, and synthesize relevant preclinical research to examine the effects of NEDD4L signaling on atherosclerosis. To ensure broad coverage, we conducted the literature search across multiple scientific databases, including PubMed, Google Scholar, Web of Science, Scopus, and Embase. A comprehensive search strategy was developed using keywords and Medical Subject Heading (MeSH) terms related to atherosclerosis and NEDD4L. Keywords such as “NEDD4L”, “atherosclerosis”, “signaling pathway”, “cardiovascular health”, “inflammation”, “diabetes”, “hypertension”, “endothelial dysfunction”, “endothelial cells”, and “lipids” were used. Boolean operators refined the search results to ensure relevance, with examples including “NEDD4L AND Atherosclerosis AND Endothelial Dysfunction” and “Atherosclerosis AND Cardiovascular Health AND NEDD4L”. The inclusion criteria comprised peer-reviewed studies and original research using in vitro and animal models. Systematic reviews and meta-analyses were excluded. Clinical trials were also excluded due to their unavailability in the current scope. Relevant studies investigated the impact of NEDD4L signaling on atherosclerosis. Articles published from January 2000 to January 2025 were included to capture recent advancements. Exclusion criteria encompassed articles that did not focus on the relationship between NEDD4L and atherosclerosis and those unrelated to this health condition. Non-English publications were excluded unless a translation was available. The initial search yielded numerous results, screened by title and abstract to determine relevance based on the inclusion criteria. Articles available only as abstracts, without full-text access, were not considered. Then, full texts were assessed for eligibility. At this stage, duplicate studies were removed, and the remaining articles were reviewed in detail. Information, including study design and methods, intervention details, outcomes related to NEDD4L signaling against atherosclerosis, and study limitations and conclusions, was extracted from each study. The quality of the included studies was assessed based on two authors’ experience (L.F.L. and S.M.B.) to evaluate aspects such as study design, methodological rigor, and relevance. The extracted data were gathered and synthesized to identify common themes, patterns, and discrepancies across studies. A narrative synthesis was employed to integrate the findings and provide a comprehensive overview of NEDD4L’s effects on atherosclerosis occurrence across different models.

### 3.2. Literature Search Report

One-hundred-twenty-five records were identified from databases during the review’s initial phase, and an additional 36 records were identified from registers. After compiling these records, those unsuitable for further consideration were discarded. This process eliminated 48 duplicate records, and 52 were marked as ineligible by automation tools; 20 records were removed for other reasons, including issues such as incomplete data or irrelevant content. After eliminating these records, 41 records remained for the screening process. During the screening phase, 32 records were removed based on their content, relevance, or quality. Nine records remained and were targeted for retrieval. All reports were successfully retrieved without any issues and were assessed for eligibility to determine if they met the criteria for inclusion in the review. At this stage, four records were excluded due to being a non-experimental paper, not involving NEDD4L signaling or being unrelated to atherosclerosis physiopathology, not being based on preclinical models, and not being in English. This resulted in five studies ultimately being included in the review. It is worth noting that no additional reports were found or associated with the studies included. To illustrate the literature search report, Figure 3 depicts the distribution of the studies across the various search stages, as detailed in the Preferred Reporting Items for Systematic Reviews and Meta-Analyses (PRISMA) flow chart.

### 3.3. Implications of Targeting NEDD4L Signaling Against Atherosclerosis: Results of the Included Studies and Future Research Directions

Using HCMECs exposed to diabetic conditions, Chen et al. [38] evaluated the effects of the secretome of the treated cells. They demonstrated that the HG and oxLDL elements exposed to those cells significantly altered their metabolomics. Within this scenario, the diseased cells started to secrete NEDD4L-rich exosomes in significant amounts. Via further biochemical and biophysical analyses, the authors highlighted the detrimental effects of NEDD4L, which significantly interfered with atherosclerosis, increasing atherosclerosis plaque formation and interfering with disease progression. The mechanisms behind these actions were mediated through the enhanced ubiquitination of IκBα and PPARγ alongside increased phosphorylation. In addition, NEDD4L upregulation significantly altered macrophage polarization, stimulating the M1 pro-inflammatory profile, oxLDL uptake stimulation, and foam cell formation. To elaborate on their findings further, Chen et al. [61] published another research article delving into the rationale behind the HG + oxLDL + NEDD4L-induced atherosclerosis axis in HCMECs. Cells were cultured with small interfering ribonucleic acid (siRNA) NEDD4L, and exosomes from ADMSCs were transfected with an overexpressed RASGRP2 vector. RASGRP2 is an essential factor contributing to endothelial cell dysfunction protection and is often utilized in atherosclerosis research due to its crucial role in cellular autophagy during oxygen deprivation. In their study, Chen et al. [61] demonstrated that NEDD4L promotes RASGRP2 ubiquitination and consequent degradation, ultimately contributing to atherosclerosis through diminished endothelial cell viability, decreased endothelial cell migration, and worsened angiogenesis.

The results of the studies above are essential since they underscore NEDD4L as a potential candidate for intervening in atherosclerosis occurrence. In this scenario, NEDD4L has the potential as a diagnostic marker of a diabetic burden on endothelial cells, highlighting modifications with an endothelial ambiance that could predispose to enhanced atherosclerosis plaque formation. In addition, the results of the studies show that disrupting NEDD4L is a candidate therapy against atherosclerosis. Therefore, further research must focus on in vivo experiments to translate the findings into well-designed clinical trials to screen human individuals and promote the pharmaceutical design of products that could target NEDD4L in atherosclerosis therapy. Using exosomes to elaborate therapies could also be a valuable strategy. Exosomes are naturally occurring extracellular vesicles generated by all cells. They can carry nucleic acids, proteins, metabolites, and lipids [79]. Although the second study by Chen et al. evaluated the effects of exosomes derived from ADMSCs on endothelial cells, since adipose tissue is the most abundant tissue within the body, future research could also introduce endothelial secretome as exosomes to ameliorate plaque targeting.

Song et al. [33] is the third study to demonstrate the deleterious effects of NEDD4L in stimulating atherosclerosis. These authors utilized high-fat diet (HFD)-induced Apolipoprotein (Apo) E ^−/−^ mice and raw macrophages transfected with NEDD4L to evaluate whether NEDD4L counteracts atherosclerotic cardiovascular disease occurrence. The results demonstrated that NEDD4L expression increased the M1/M2 macrophage ratio and oxLDL uptake due to increased PPARγ ubiquitination and SMAD phosphorylation. On the other hand, NEDD4L genetic inhibition significantly diminished pro-inflammatory factor production and alleviated the atherosclerotic process. These results are promising since they demonstrated that counteracting NEDD4L signaling could be effective against atherosclerosis, possibly diminishing atherosclerotic lesions and inflammatory signaling in human patients. Unlike statins, which block cholesterol synthesis and represent the primary intervention against atherosclerosis nowadays, Song et al. [33] investigated whether genetic therapy could influence atherosclerosis plaque formation. Using genetic interventions against cardiovascular disorders is a significant strategy since it opens doors to personalized treatment strategies, which are paramount in customized medicine due to their precision [80]. Figure 4 demonstrates some implications of targeting NEDD4L signaling against atherosclerosis based on Chen et al.’s [38,61] and Song et al.’s studies [33].

On the contrary, Liu et al. [53] demonstrated that NEDD4L might have a positive role in mitigating atherosclerosis through its signaling. These authors treated HUVECs transfected with NEDD4L-siRNA and infected with NEDD4L-adenovirus to evaluate whether NEDD4L could regulate endothelial cell function to counteract atherosclerosis plaque formation. Their results demonstrated that NEDD4L promotes Akt, ERK 1/2, and eNOS phosphorylation, VEGFR2 expression, and angiogenesis, ultimately leading to endothelial cell migration and improved function, counteracting atherosclerosis. Due to these effects, NEDD4L signaling may be a promising strategy to treat and decrease the progression of some ischemic diseases, such as those related to heart infarction. Since atherosclerosis is also a prominent pathway in peripheral artery diseases, stimulating NEDD4L signaling in these patients may also be an effective strategy despite the current advancements in the field [81].

Lastly, Gao et al. [78] applied the SHENQI compound (composed of several medicinal plants, including *Rehmannia glutinosa*, hawthorn, raw radix, and ginseng) from traditional Chinese medicine to a mouse model of diabetic microangiopathy to evaluate whether NEDD4L targeting could protect against vascular calcification during diabetic angiopathy. The results were dubious. The SHENQI compound was found to target NEDD4L signaling, but the specific molecular mechanisms involved were not elucidated. Additionally, whether the compound could up- or downregulate NEDD4L signaling was not assessed. Although it may offer a new strategy to counteract atherosclerosis via medicinal plants and bioactive compounds targeting NEDD4L under diabetic conditions, further studies must be conducted to evaluate whether SHENQI intervenes directly in diabetes associated with NEDD4L.

The broader medical community may be most interested in targeting NEDD4L and atherosclerosis using medicinal plants and bioactive compounds. These are often more cost-effective, present few adverse events, and are highly effective against cardiovascular diseases and outcomes [82]. Introducing medicinal plants and bioactive compounds into cardiovascular disease treatment could offer other promising benefits since it aligns with green healthcare principles. Green healthcare emphasizes environmentally friendly practices, exemplified using natural substances. Incorporating such biocompatible and sustainable therapies into cardiovascular therapies respects environmental considerations and can reduce the reliance on synthetic chemicals [83].

## 4. Conclusions

Our analysis of the included studies underscores that NEDD4L signaling may have heterogeneous effects on atherosclerosis. Although most included studies underscored its deleterious potential, increasing atherosclerosis plaque formation, one study indicated that stimulating NEDD4L may positively counter atherosclerosis plaque formation. However, the results from that study were dubious and not directly related to NEDD4L signaling since the authors used bioactive compounds to treat endothelial cells in a mouse model of diabetic angiopathy. The beneficial effects might be associated with antidiabetic actions and not directly with NEDD4L modulation. Therefore, future research endeavors must address several limitations. These critical limitations have been tentatively highlighted throughout this manuscript to aid more detailed, informative, and vital future research initiatives based on preclinical studies to successfully translate the findings into clinical, randomized, and masked trials. Figure 5 revisits our study’s rationale and responds to the questions raised within Figure 1 after our complete evaluation of the current literature on NEDD4L signaling in atherosclerosis.

Regarding clinical applicability, it is worth noting that NEDD4L-targeted therapies could be integrated synergistically into existing treatment options and strategies against atherosclerosis, including statins, anti-inflammatory agents, and lifestyle modifications. Combining NEDD4L-targeted therapies with these existing treatment options could be efficient due to various effects. Statins are cholesterol-lower prescribed drugs that reduce cardiovascular event occurrence [84]. In contrast, NEDD4L influences oxLDL uptake by polarized macrophages in blood vessels, preventing foam cell formation and atherosclerosis [38]. Anti-inflammatory drugs are also a class prominently related to NEDD4L-targeted therapies since anti-inflammatory agents can treat atherosclerosis-related inflammation [85]. NEDD4L-targeted therapies could enhance the anti-inflammatory potential of these drugs, therefore reducing pro-inflammatory cytokines more effectively and preventing macrophage polarization to the inflammatory pathway, leading to better atherosclerosis treatment results. Dietetic, exercise, and smoking interventions are crucial in targeting atherosclerosis [86,87]. In this scenario, NEDD4L-targeted therapies may be essential due to their anti-inflammatory, lipid-lowering, and antioxidant effects, predisposing individuals to more pronounced effects and influencing their acceptance of lifestyle interventions, leading to more rapid and straightforward results.

The main point of discussion from the included studies is whether NEDD4L’s effects on angiogenesis and endothelial function are detrimental to atherosclerosis. The included studies are confusing since some point to benefits and others to harm caused by NEDD4L signaling concerning the abovementioned effects. It is worth noting that enhanced blood flow due to the pro-angiogenic impacts effectively promotes new blood vessel formation, enhancing endothelial function and improving blood flow to ischemic tissues, including those with atherosclerosis, which supports tissue repair and regeneration. In contrast, the blood supply is compromised [88,89,90,91]. However, increased angiogenesis may also lead to the formation of fragile blood vessels within atherosclerosis plaques, which compromise their structure and lead to ruptures and acute cardiovascular events such as heart attacks and strokes [92,93]. Beyond this, enhanced angiogenesis may also promote inflammation, exacerbating atherosclerosis progression and contributing to further vascular damage [94]. In summary, while targeting NEDD4L may improve blood supply and tissue regeneration under lower blood flow, the risks associated with its use for promoting angiogenesis may not surpass the benefits due to its associations with atherosclerosis plaque stability and potential rupture. Future clinical research must elucidate the findings and apply NEDD4L therapy in predisposed individuals, including personalized medicine based on their metabolic, immunologic, and genetic profiles.

## Figures and Tables

**Figure 1 biology-14-00220-f001:**
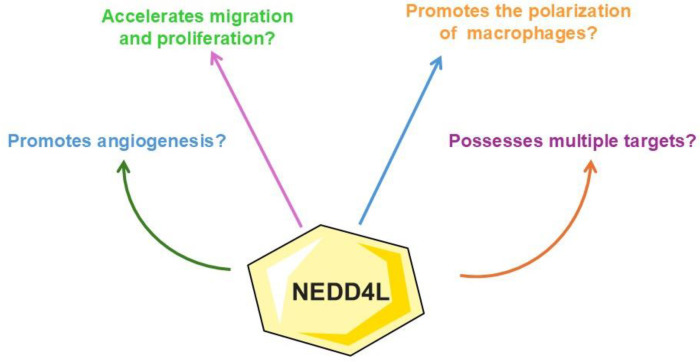
Study rationale. NEDD4L may have several potential effects against atherosclerosis. These effects are mainly related to angiogenesis, migration, proliferation of endothelial cells, and macrophage polarization. However, many signaling pathways might be affected, so multiple targets could be possible. The present review aims to address a gap in the current literature, provide results from the evidence gathered in this field, and propose future research avenues. Abbreviations: NEDD4L, Neural Precursor Cell-Expressed Developmentally Downregulated 4-Like.

**Figure 2 biology-14-00220-f002:**
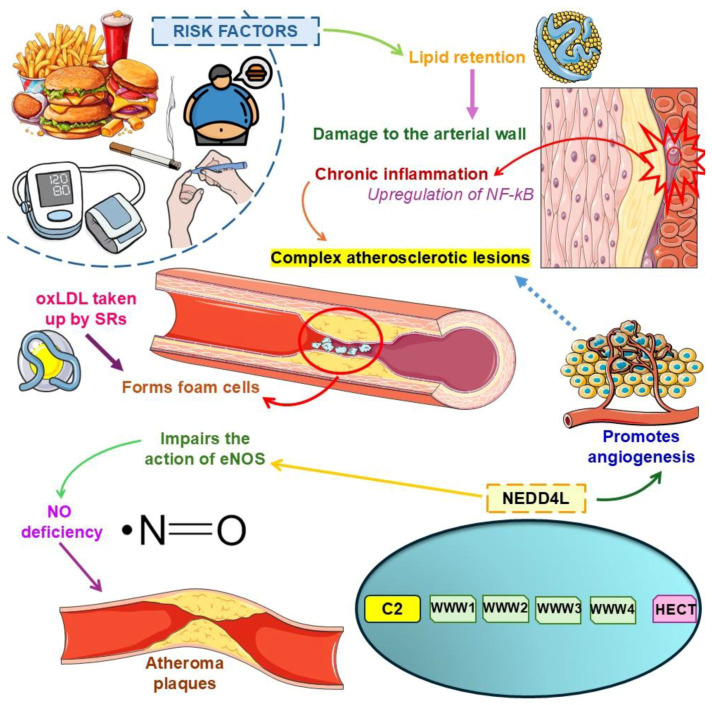
Atherosclerosis physiopathology with emphasis on NEDD4L signaling. Abbreviations: eNOS, Endothelial Nitric Oxide Synthase; HECT, Homologous to E6-AP C-Terminus; NEDD4L, Neural Precursor Cell Expressed Developmentally Downregulated 4-Like; NF-κB, Nuclear Factor Kappa B; NO, Nitric Oxide; oxLDL, Oxidized Low-Density Lipoprotein; SRs, Scavenger Receptors.

**Figure 3 biology-14-00220-f003:**
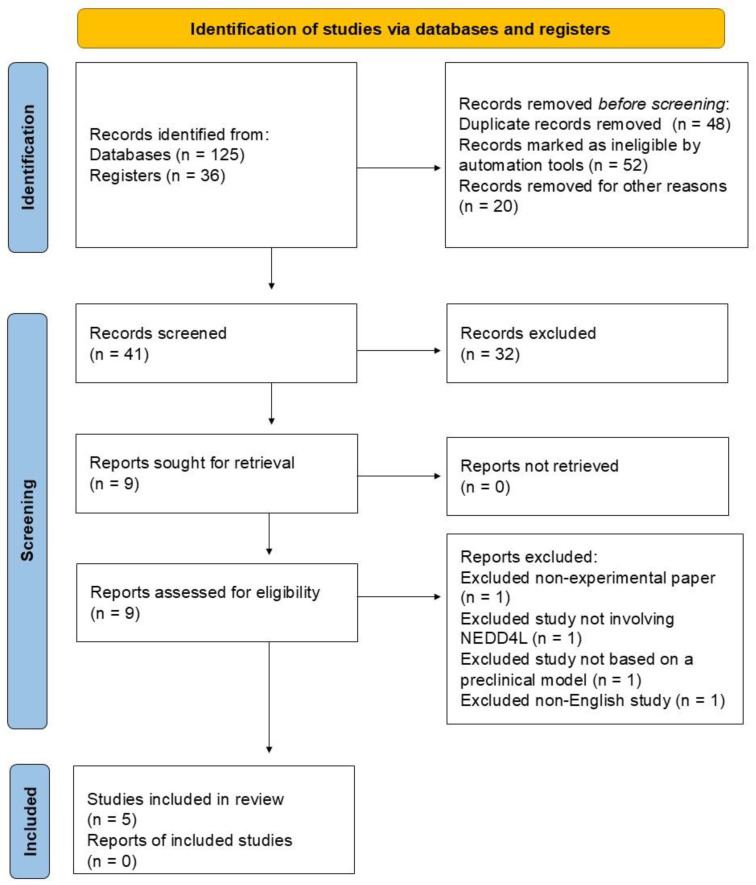
Distribution of studies by search stage and focus area: PRISMA flow chart overview.

**Figure 4 biology-14-00220-f004:**
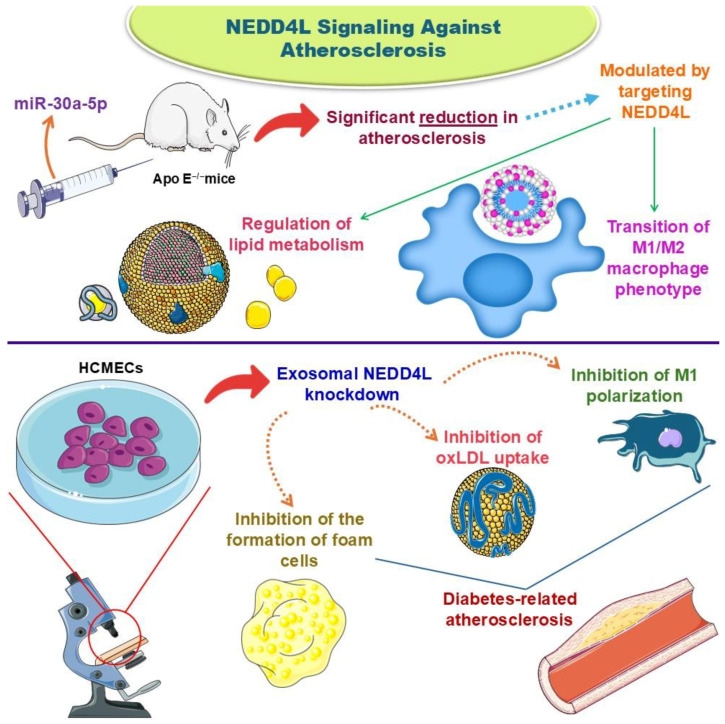
Implications of targeting NEDD4L signaling against atherosclerosis based on Chen et al.’s [38,61] and Song et al.’s studies [33]. Abbreviations: Apo, Apolipoprotein; HCMECs, Human Cardiac Microvascular Endothelial Cells; miR, Micro Ribonucleic Acid; NEDD4L, Neural Precursor Cell Expressed Developmentally Downregulated 4-Like; oxLDL, Oxidized Low-Density Lipoprotein.

**Figure 5 biology-14-00220-f005:**
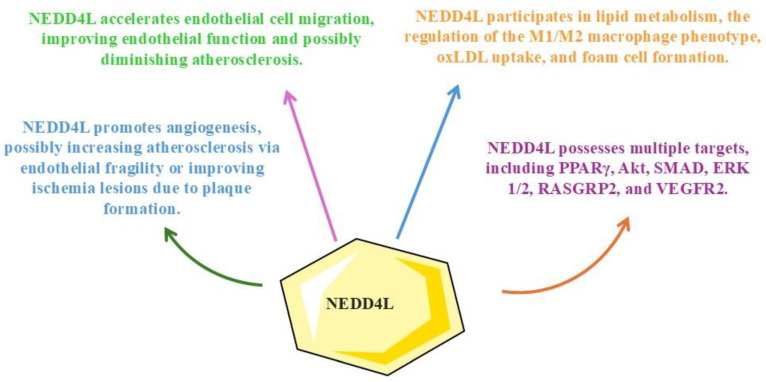
Revisiting our study’s rationale and responding to the questions raised regarding NEDD4L signaling against atherosclerosis: implications for the buildup of fats. Abbreviations: Akt, Protein Kinase b; ERK, Extracellular Signal-Regulated Kinase; NEDD4L, Neural Precursor Cell Expressed Developmentally Downregulated 4-Like; oxLDL, Oxidized Low-Density-Lipoprotein; PPARγ, Peroxisome Proliferator-Activated Receptor Gamma; RASGRP2, Ras Guanyl-Releasing Protein 2; SMAD, Suppressor of Mothers Against Decapentaplegic; VEGFR2, Vascular Endothelial Growth Factor Receptor 2.

**Table 1 biology-14-00220-t001:** Targeting atherosclerosis via NEDD4L signaling analyzing the implications for atherosclerosis treatment and progression.

Ref	Model	Study Aims	Molecular Mechanisms and Outcomes	Possible Clinical Applications	Future Research Endeavors
[38]	HCMECs exposed to HG and oxLDL.	Investigate whether exosomes containing NEDD4L derived from these cells could negatively interfere with disease progression.	⬆ IκBα and PPARγ ubiquitination, ⬆ SMAD phosphorylation. In HG + oxLDL-induced HCMECs, NEDD4L is overexpressed, and it promotes ⬆ macrophage M1 polarization, ⬆ oxLDL uptake, and ⬆ foam cell formation.	Targeted therapies to decrease NEDD4L expression in endothelial cells could be developed to mitigate damage caused by cardiovascular diseases.	In vivo experiments are necessary for the translational use of NEDD4L in screening and treating cardiovascular disorders.
[61]	HG + oxLDL-induced HCMECs co-cultured with siRNA NEDD4L and ADMSCs exosomes transfected with RASGRP2 overexpression vector.	Evaluate the role of the NEDD4L/RASGRP2 axis in DM-related atherosclerosis progression.	⬆ RASGRP2 ubiquitination and degradation, ⬇ RASGRP2’s protective effects against atherosclerosis, ⬇ cell viability, ⬇ cell migration, ⬇ cell angiogenesis, ⬆cell permeability, and ⬆ ROS production.	Drugs targeting the knockdown of NEDD4L could be a valuable alternative to combat cardiovascular diseases.	Strategies incorporating NEDD4L signaling must apply to targeted DM-associated atherosclerosis therapy as a first step in animal investigations.
[33]	HFD-induced Apo E ^−/−^ mice treated with ago-miR-30a-5p and macrophages transfected with miR-30a-5p or NEDD4L siRNA.	Demonstrate whether miR-30a-5p interacts with NEDD4L to attenuate atherosclerosis.	⬆ M1/M2 ratio, ⬆ oxLDL uptake, ⬆ PPARγ ubiquitination, ⬆ SMAD phosphorylation. miR-30a-5p inhibits NEDD4L, leading to an ⬆ in anti-inflammatory cytokines and a ⬇ in pro-inflammatory factors.	Counteracting the NEDD4L pathway could be valuable in reducing the formation of atherosclerotic lesions and inflammatory responses.	Additional animal experiments are paramount to assess miR-30a-5p and NEDD4L as targets in atherosclerosis management.
[53]	HUVECs transfected with NEDD4L-siRNA and infected with NEDD4L-adenovirus.	Evaluate NEDD4L as an endothelial cell function regulator.	⬆ Akt, ⬆ ERK 1/2 and ⬆ eNOS phosphorylation, ⬆ VEGFR2, ⬆ cell cycle-related proteins cyclin D1 and D3, ⬆ angiogenesis, ⬆ cell proliferation, ⬆ cell migration, ⬆ endothelial function, ⬇ hypertension and atherosclerosis pathology.	NEDD4L emerges as a therapeutic alternative for treating ischemic diseases directly associated with atherosclerosis.	NEDD4L-mediated angiogenesis should be analyzed under several pathological conditions to explore the molecular pathways.
[78]	The SHENQI compound was administered to a diabetic model.	Identify NEDD4L transcription factors that may cause diabetic vascular damage.	The SHENQI compound interacts with the NEDD4L signaling.	NEDD4L may offer new and comprehensive therapeutic strategies against DM and atherosclerosis-predominant angiopathy.	More preclinical studies should validate the SHENQI compound’s safety and effectiveness while targeting NEDD4L.

Abbreviations: ⬆, increase; ⬇, decrease; ADMSCs, Adipose-Derived Mesenchymal Stem Cells; Akt, Protein Kinase b; Apo, Apolipoprotein; DM, Diabetes Mellitus; eNOS, Endothelial Nitric Oxide Synthase; ERK, Extracellular Signal-Regulated Kinase; HG, High Glucose; HCMECs, Human Cardiac Microvascular Endothelial Cells; HFD, High-Fat Diet; HUVECs, Human Umbilical Vein Endothelial Cells; IκBα, Nuclear Factor of Kappa Light Polypeptide Gene Enhancer in B-cells Inhibitor Alpha; miR, Micro Ribonucleic acid; NEDD4L, Neural Precursor Cell Expressed Developmentally Downregulated 4-Like; oxLDL, Oxidized Low-Density Lipoprotein; PPARγ, Peroxisome Proliferator-Activated Receptor Gamma; RASGRP2, Ras Guanyl-Releasing Protein 2; SMAD, Suppressor of Mothers Against Decapentaplegic; siRNA, Small Interfering Ribonucleic Acid; VEGFR2, Vascular Endothelial Growth Factor Receptor 2; ROS, Reactive Oxygen Species.

## Data Availability

No new data were created or analyzed in this study. Data sharing is not applicable to this article.

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
