# Peer review of "Targeting Atherosclerosis via NEDD4L Signaling—A Review of the Current Literature"

_biology, 2025, doi:10.3390/biology14030220_

Round 1
Reviewer 1 Report
Comments and Suggestions for Authors
The manuscript is well-written and addresses an important topic, but it requires Major revisions to meet the standards of the journal. I recommend that the manuscript be reconsidered for publication after these revisions have been made.
1. The current title, "Targeting Atherosclerosis via the NEDD4L Signaling – Molecular Implications Against These Buildups of Fats: A Review of the Current Literature," is somewhat lengthy and could be confusing. Consider revising it to be more concise.
2. Abstract lacks specific details about the methodology used for the literature search. Including information about the databases searched (e.g., PubMed, Scopus), inclusion/exclusion criteria, and the timeframe of the search would enhance the review.
3. In the Introduction pay more focused discussion on the specific gap in the literature. Specifically, the authors should emphasize why NEDD4L has not been thoroughly reviewed in the context of atherosclerosis despite its known involvement in cardiovascular diseases.
4. Overview of Atherosclerosis Physiopathology: This section is somewhat lengthy and could be streamlined. For example, the detailed discussion on macrophage polarization (M1/M2) is informative but somewhat tangential to the main focus on NEDD4L. Consider summarizing this part and focusing more on how NEDD4L interacts with these pathways.
5. For (NEDD4L's Role in Angiogenesis and Endothelial Function) the authors should clarify whether NEDD4L's pro-angiogenic effects are beneficial or detrimental in the context of atherosclerosis, as this is a point of potential confusion.
6. The inclusion of only five studies is limited, especially given the broad scope of the review. The authors should justify why so few studies were included.
7. Discuss the potential clinical applications of NEDD4L-targeted therapies in more detail, including how they might be integrated into current treatment paradigms.
8. Table 1 needs to improve and make it more presentable.
Comments on the Quality of English Language1. There are minor inconsistencies in terminology. For example, "atheroma plaque" and "atherosclerotic plaque" are used interchangeably. Authors should Standardized terms throughout the manuscript.
2. Address minor grammatical issues
Author Response
RESPONSE TO REVIEWERS' COMMENTS
Manuscript number: biology-3469525 † Biology (MDPI)
"Targeting Atherosclerosis via the NEDD4L Signaling – A Review of the Current Literature"
The authors of this document wish to express their deepest gratitude to the Editor-in-Chief and the Reviewer for their thorough and insightful evaluation of our manuscript. Their expert feedback has been invaluable in enhancing the quality of our work. We have carefully considered and diligently implemented each suggestion, significantly improving the manuscript. We have made substantial revisions to address the points raised. These noteworthy changes are marked mainly with YELLOW-highlighted text throughout the document for ease of reference. For corrections highlighted with a different color, there will be a note for the referee advertising. Additionally, we have prepared a detailed and comprehensive response to each comment and suggestion. This response is organized in a "point-by-point" format below, ensuring that every concern has been thoroughly addressed and explained. We sincerely appreciate the time and effort invested by the Editor-in-Chief and the Reviewer, and we believe their contributions have significantly strengthened the final version of our manuscript.
General comment
The manuscript is well-written and addresses an important topic, but it requires Major revisions to meet the standards of the journal. I recommend that the manuscript be reconsidered for publication after these revisions have been made.
Response
Dear Erudite Reviewer, thank you for taking the time to revise our manuscript and allowing us to improve based on your precious comments and suggestions. After addressing all your comments and suggestions regarding our manuscript text, we are confident that a significantly enhanced manuscript version has emerged. We are excited to resubmit the modified version for your perusal and reevaluation. Thank you for your brilliant insights, essential contributions, and feedback. You do have an eye for improvement. As a signal of our utmost respect for you, we want to provide you with a detailed and comprehensive point-by-point response to your comments below. Thank you once again for your time and patience in revising our article.
Comment #1
The current title, "Targeting Atherosclerosis via the NEDD4L Signaling – Molecular Implications Against These Buildups of Fats: A Review of the Current Literature," is somewhat lengthy and could be confusing. Consider revising it to be more concise.
Response
Dear Erudite Reviewer, thank you for this insightful suggestion and attention to improvement. You are correct, and we agree that correcting our title would enhance its quality, readability, and understanding. Therefore, to fix our manuscript, we have modified the title, which is now presented correctly within Lines 1-3 on Page 1 of the revised manuscript’s document. To facilitate your kind review, we -also show the corrected title below for your convenience, given your utmost importance in assessing our manuscript. The new title is "Targeting Atherosclerosis via the NEDD4L Signaling – A Review of the Current Literature.” Again, thank you for your commitment to improving our manuscript. We are sure our text has significantly improved after we addressed your precious input.
Comment #2
Abstract lacks specific details about the methodology used for the literature search. Including information about the databases searched (e.g., PubMed, Scopus), inclusion/exclusion criteria, and the timeframe of the search would enhance the review.
Response
Dear Erudite Reviewer, you are entirely correct, and we agree with you that adding more methodological details to our abstract would undoubtedly enhance its quality. Thank you for your attention to detail and eye for improvement. To correct our manuscript accordingly, we added Lines 44-53 on Pages 1-2 of the revised manuscript document, delving into methodological details, including the databases searched, inclusion and exclusion criteria, and search timeframe. Our databases were PubMed, Google Scholar, Web of Science, Scopus, and Embase. Our inclusion criteria comprised peer-reviewed original studies using in vitro and animal models due to the unavailability of relevant clinical studies. Our exclusion criteria comprised systematic reviews, meta-analyses, and articles that did not focus on the relationship between NEDD4L and atherosclerosis of those unrelated to this health condition. We selected studies from January 2000 to January 2025 to capture recent advancements.
Again, thank you for your patience and guidance. Your contributions have been instrumental in reshaping our manuscript. We are proud of the revised version we are presenting to you, and we eagerly anticipate your positive response and approval for its publication.
Comment #3
In the Introduction pay more focused discussion on the specific gap in the literature. Specifically, the authors should emphasize why NEDD4L has not been thoroughly reviewed in the context of atherosclerosis despite its known involvement in cardiovascular diseases.
Response
Dear Erudite Reviewer, thank you for this insightful and essential suggestion. You are correct, and we agree that we must emphasize our research topic with the utmost transparency to achieve our goals for a complete and critical publication. To improve our manuscript accordingly, we included Lines 116-123 on Page 3 of the revised manuscript’s document. In these lines, we delved into the novelty of our review and the reasons for its necessity in the light of emergency evidence. NEDD4L is significantly linked to the occurrence of cardiovascular diseases, but the evidence connecting NEDD4L signaling to atherosclerosis is relatively new. Until 2024, when three recent studies were published, the information available on NEDD4L and its role in atherosclerosis was limited. Considering these discoveries and the existing research gaps, the importance of this urgent area of study has been renewed. As a result, a systematic review of this topic has become essential.
Again, thank you for your interest in our work and for assessing it with the utmost criteria. We are proud to present you with the revised version, and we eagerly anticipate a positive response from you. It is a true honor to communicate with such an esteemed reviewer. Thank you for everything!
Comment #4
Overview of Atherosclerosis Physiopathology: This section is somewhat lengthy and could be streamlined. For example, the detailed discussion on macrophage polarization (M1/M2) is informative but somewhat tangential to the main focus on NEDD4L. Consider summarizing this part and focusing more on how NEDD4L interacts with these pathways.
Response
Dear Erudite Reviewer, thank you for this suggestion. We appreciate your attention to detail, eye for improvement, and guidance. You are entirely correct, and we agree that Subsection “2.3. An Overview of Atherosclerosis Physiopathology with Emphasis on NEDD4L Signaling” may be too lengthy and would benefit from summarization. Therefore, we implemented modifications throughout the subsection, which you can find in Lines 225-298 on Pages 5-7, highlighted using YELLOW-highlighted text. I want to bring to your attention the revised Subsection 2.3. now spans into 73 lines, representing a decreased 32 lines (previously, the subsection spanned 105 lines). We made the necessary corrections within all subsection’s paragraphs. Mainly, I would like to bring to your attention the modifications we made on Lines 237-243 on Page 6 of the revised document, delving significantly into NEDD4L signaling in macrophage polarization following the study by Song et al., which is included in this section. After addressing your corrections, we focused more on NEDD4L signaling within this paragraph, and its length decreased from 17 lines (previous Lines 216-233) to 6 in total.
Again, thank you for your precious input, time, and consideration. Your contributions have been invaluable in enhancing our manuscript’s quality and readability, and we are proud to present you with this revised version. Thank you for your guidance.
Comment #5
For (NEDD4L's Role in Angiogenesis and Endothelial Function) the authors should clarify whether NEDD4L's pro-angiogenic effects are beneficial or detrimental in the context of atherosclerosis, as this is a point of potential confusion.
Response
Dear Erudite Reviewer, thank you for this important suggestion. You are entirely correct, and we agree that summarizing the controversial findings in our manuscript would significantly benefit its overall quality and readability. Therefore, we included Lines 498-515 on Page 15, delving into the rationale behind using NEDD4L signaling in promoting angiogenesis and modifying endothelial function. In summary, targeting NEDD4L may enhance blood supply and promote tissue regeneration under conditions of lower blood flow. However, the risks associated with its role in angiogenesis might outweigh the benefits, given their links to atherosclerosis plaque stability and the potential for plaque rupture. Future clinical research is essential to clarify these findings and to investigate the application of NEDD4L therapy in individuals who are predisposed to vascular issues. This research should also incorporate personalized medicine approaches considering metabolic, immunologic, and genetic profiles. Thank you for your guidance, patience, and understanding. Your attention to detail and eye for improvement have undoubtedly shaped our manuscript. We eagerly anticipate your positive response and the approval of our revised version for publication in this critical journal.
Comment #6
The inclusion of only five studies is limited, especially given the broad scope of the review. The authors should justify why so few studies were included.
Response
Dear Erudite Reviewer, thank you for this comment and for the opportunity to clarify our included studies. NEDD4L is closely related to the occurrence of cardiovascular diseases, but the evidence linking NEDD4L signaling to atherosclerosis is primarily new. Until 2024, when three contemporary studies were published, there was a lack of comprehensive detail regarding the role of NEDD4L in atherosclerosis. Due to this novelty, we highlighted in our manuscript the included studies extensively and promoted their analyses in both in-text, in-table, and in-figure models. In addition, due to the novelty of the issue, previous review articles have not significantly provided evidence regarding NEDD4L signaling on atherosclerosis. Given these recent discoveries and the existing research gaps, the significance of this emerging field of study has been revitalized, highlighting the necessity for a systematic review of this topic. Additionally, we want to highlight that although the included studies are five, their evidence is tremendous. The included studies converge on many points. However, they are particularly interesting due to their divergent points. For example, the comment above delves into the dubious actions of NEDD4L on angiogenesis and endothelial function, improving blood supply to organs but harming atherosclerosis.
Finally, we obeyed a rigorous methodology, encompassing only studies related to NEDD4L and atherosclerosis, giving us a specific focus to fill a particular gap in the literature. Beyond this, the manuscript also discusses evidence underscoring the broader evidence on NEDD4L signaling in cardiovascular diseases, although more briefly than in the case of atherosclerosis.
Thank you for clarifying these points with us. We are truly thankful for the opportunity to communicate with such a critical and esteemed reviewer. Thank you for your consideration and for assessing our manuscript’s suitability for publication in this vital journal.
Comment #7
Discuss the potential clinical applications of NEDD4L-targeted therapies in more detail, including how they might be integrated into current treatment paradigms.
Response
Dear Erudite Reviewer, thank you for this essential suggestion. You are entirely correct, and we agree that delving into the detailed potential clinical applications of NEDD4L-targeted therapies during daily clinical practice would undoubtedly enhance our manuscript’s analyses. Therefore, we implemented Lines 481-497 on Page 14 to correct our manuscript accordingly to delve into the abovementioned applicability. NEDD4L-targeted therapies could be effectively integrated with existing treatments for atherosclerosis, such as statins, anti-inflammatory agents, and lifestyle changes. Statins lower cholesterol and reduce cardiovascular events, while NEDD4L influences the uptake of oxLDL in macrophages, preventing foam cell formation. Anti-inflammatory drugs can work alongside NEDD4L-targeted therapies to reduce inflammation and prevent macrophage polarization, leading to better treatment outcomes. Additionally, NEDD4L-targeted therapies may enhance the effects of diet, exercise, and smoking cessation, thanks to their anti-inflammatory, lipid-lowering, and antioxidant properties, resulting in quicker and more significant improvements.
Comment #8
Table 1 needs to improve and make it more presentable.
Response
Dear Erudite Reviewer, thank you for bringing this to our attention. You have an eye for detail and improvement. To correct our manuscript accordingly, we made amendments to Table 1 to make it more presentable. We have modified the spacing between lines using Microsoft Word software to 1.0 and the font size to 10, respecting MDPI’s standards. Additionally, we implemented modifications, so Table 1 is initialized together with its title from the rest of the text, therefore improving its readability and presentation of content. Please refer to Pages 8-9 for the revised manuscript’s Table 1. Thank you for your guidance. Your contributions have been instrumental in reshaping our manuscript for the better. We are eagerly anticipating a positive response from you and the acceptance of our manuscript for publication in this important journal.
Comment #9
There are minor inconsistencies in terminology. For example, "atheroma plaque" and "atherosclerotic plaque" are used interchangeably. Authors should Standardized terms throughout the manuscript.
Response
Dear Erudite Reviewer, thank you for this important comment. You are entirely correct, and we agree that standardizing our manuscript’s terms would undoubtedly enhance its quality and readability. Therefore, we implemented modifications throughout the entire manuscript, and we are proud to present you with this revised version. As an example of our commitment to correct our manuscript based on your suggestions, please refer to the previous terms “atheroma” and/or “atherosclerotic plaques” as “atherosclerosis plaques” within the revised manuscript’s document. Since these modifications are separated from the main modifications made to the text, we highlighted those changes using BLUE-highlighted text to facilitate your kind review throughout the entire document. Thank you for your cooperation and consideration. Your review has shaped our manuscript for the better. And we are eagerly anticipating a positive response from you at the revised version. Thank you for everything!
Comment #10
Address minor grammatical issues
Response
Dear Erudite Reviewer, thank you for this suggestion. We appreciate your commitment to improving our manuscript’s readability. Therefore, I want to share that our English-native speakers have diligently revised the manuscript for minor grammatical typos and issues. We hope our manuscript meets your highest standards now. We appreciate your help throughout the peer-review process, and we are thankful for the opportunity to communicate with such a critical and esteemed reviewer. Thank you for everything!
I, the corresponding author of the manuscript "Targeting Atherosclerosis via the NEDD4L Signaling – A Review of the Current Literature" under the assigned ID biology-3469525, on behalf of my coauthors, once again extend my heartfelt gratitude to the knowledgeable Editor-in-Chief and reviewers for their time and expertise in revising our manuscript. After we addressed their constructive and refined feedback and suggestions, a significantly improved manuscript version emerged. Undoubtedly, their insightful suggestions and feedback have significantly enhanced the quality of our manuscript. We respectfully are at the disposal of the Editor-in-Chief and the Reviewer to address any additional suggestions regarding our publication. Suppose you are satisfied with our newly refined and significantly improved version. In that case, we are eager to anticipate the acceptance of our article for publication in this most critical journal, Biology. Thank you once again for your time and expertise.
Reviewer 2 Report
Comments and Suggestions for Authors
Thank you for efforts in writing a comprehensive review on the role of NEDD4L signaling in atherosclerosis. this will be very helpful for researchers who has interest in diving into this topic. Section 2.3, in my opinion, seems to be the most important part of this review. A good amount of literatures were carefully analyzed, but this section needs some revisions. I hope some of the suggestions will improve the manuscript.

Author Response
RESPONSE TO REVIEWERS' COMMENTS
Manuscript number: biology-3469525 † Biology (MDPI)
"Targeting Atherosclerosis via the NEDD4L Signaling – A Review of the Current Literature"
The authors of this document wish to express their deepest gratitude to the Editor-in-Chief and the Reviewer for their thorough and insightful evaluation of our manuscript. Their expert feedback has been invaluable in enhancing the quality of our work. We have carefully considered and diligently implemented each suggestion, significantly improving the manuscript. We have made substantial revisions to address the points raised. These noteworthy changes are marked mainly with YELLOW-highlighted text throughout the document for ease of reference. For corrections highlighted with a different color, there will be a note for the referee advertising. Additionally, we have prepared a detailed and comprehensive response to each comment and suggestion. This response is organized in a "point-by-point" format below, ensuring that every concern has been thoroughly addressed and explained. We sincerely appreciate the time and effort invested by the Editor-in-Chief and the Reviewer, and we believe their contributions have significantly strengthened the final version of our manuscript.
General comment
Thank you for efforts in writing a comprehensive review on the role of NEDD4L signaling in atherosclerosis. this will be very helpful for researchers who has interest in diving into this topic. Section 2.3, in my opinion, seems to be the most important part of this review. A good amount of literatures were carefully analyzed, but this section needs some revisions. I hope some of the suggestions will improve the manuscript.
Response
Dear Erudite Reviewer, thank you for taking the time to revise our manuscript and allowing us to improve based on your precious comments and suggestions. After addressing all your comments and suggestions regarding our manuscript text, we are confident that a significantly enhanced manuscript version has emerged. We are excited to resubmit the modified version for your perusal and reevaluation. Thank you for your brilliant insights, essential contributions, and feedback. You do have an eye for improvement. As a signal of our utmost respect for you, we want to provide you with a detailed and comprehensive point-by-point response to your comments below. Thank you once again for your time and patience in revising our article.
Comment #1
Figure 1 could be simpler; it is unnecessarily huge. And the figure legend could be extended.
Response
Dear Erudite Reviewer, thank you for this insightful comment and brilliant suggestion. We agree with you that Figure 1 may be too lengthy, and we also agree that expanding its legend while limiting its size would undoubtedly enhance our manuscript`s quality and readability. Therefore, we modified Figure 1, which is revised on Page 3. Additionally, we expanded Figure 1`s legend to correct our manuscript based on your guidelines. Please refer to Lines 128-132 on Page 3 for the revised legend.
Again, thank you for your patience and guidance. Your contributions have been instrumental in reshaping our manuscript. We are proud of the revised version we are presenting to you, and we eagerly anticipate your positive response and approval for its publication.
Comment #2
Point 1: 1st and 2nd paragraph in section 2.3
Do NEDD4 (overexpression is beneficial) and NEDD4L (overexpression is detrimental) play opposite role in HUVECs If so, this needs to be directly stated. Maybe combine 1st and 2nd paragraphs into one paragraph. Is APEX1 relevant here If so, how?
Response
Dear Erudite Reviewer, thank you for this critical comment. You are entirely correct, and we agree that adding information about the opposite effects of NEDD4 and NEDD4L directly on our manuscript would undoubtedly enhance its overall quality and readability since we would not rely on the readers’ interpretation. We merged the first two paragraphs from Lines 226-236 on Pages 5-6 and added the requested sentence stating that NEDD4 and NEDD4L are both ligases. However, they have opposing effects on endothelial cells. NEDD4 reduces the development of atherosclerosis by decreasing inflammation and endothelial cell dysfunction, whereas NEDD4L influences endothelial cell migration, angiogenesis, and proliferation, thereby contributing to the development of atherosclerosis. Additionally, APEX1 is irrelevant. Therefore, we deleted information about this compound.
Your attention to detail and eye for improvement have undoubtedly shaped our manuscript. We eagerly anticipate your positive response and the approval of our revised version for publication in this critical journal.
Comment #3
Point 2: 3rd paragraph and 6th paragraph in section 2.3
The authors could summarize what is most important and relevant in respect to NEDD4L signaling and its relationship with atherosclerosis and omit unnecessary experimental details. For instance, in the 3rd paragraph: Only in the final sentence, NEDD4L was mentioned.
Response
Dear Erudite Reviewer, thank you for this important suggestion. You are entirely correct, and we agree with you. Therefore, we implemented modifications in Subsection 2.3. completely (Lines 225-298 on Pages 5-7), but mainly in the paragraphs mentioning the studies by Song et al. and Chen et al. In the revised document, all paragraphs have been extensively modified to only contain essential information on NEDD4L signaling in atherosclerosis. In addition, irrelevant information has been removed, especially from the new paragraphs about Song et al. in Lines 237-243 on Page 6 and Chen et al. in Lines 256-266 on Page 6. We paid special attention to omitting irrelevant experimental details from the included studies in the above-mentioned paragraphs and the whole subsection.
Again, thank you for your attention to detail and eye for improvement. Your contributions have been instrumental in reshaping our manuscript for the better. We are proud to present you with the revised version, and we eagerly anticipate a positive response. Thank you for everything!
I, the corresponding author of the manuscript "Targeting Atherosclerosis via the NEDD4L Signaling – A Review of the Current Literature" under the assigned ID biology-3469525, on behalf of my coauthors, once again extend my heartfelt gratitude to the knowledgeable Editor-in-Chief and reviewers for their time and expertise in revising our manuscript. After we addressed their constructive and refined feedback and suggestions, a significantly improved manuscript version emerged. Undoubtedly, their insightful suggestions and feedback have significantly enhanced the quality of our manuscript. We respectfully are at the disposal of the Editor-in-Chief and the Reviewer to address any additional suggestions regarding our publication. Suppose you are satisfied with our newly refined and significantly improved version. In that case, we are eager to anticipate the acceptance of our article for publication in this most critical journal, Biology. Thank you once again for your time and expertise.
Round 2
Reviewer 1 Report
Comments and Suggestions for Authors
No further comments.
Author Response
RESPONSE TO REVIEWERS' COMMENTS
Manuscript number: biology-3469525 † Biology (MDPI)
"Targeting Atherosclerosis via the NEDD4L Signaling – A Review of the Current Literature"
REVIEWER #1
General comment
No further comments.
General Response
Dear Esteemed Reviewer, thank you for taking the time to review our manuscript and for providing invaluable comments and suggestions. We have carefully addressed all your feedback and are grateful that our revised version of the manuscript was significantly improved. We are excited about the acceptance of our manuscript for publication in this critical journal. Your insights and contributions were instrumental in this process, and we genuinely appreciate your attention to detail. Thank you once again for your time and patience in reviewing our article.
Best regards,
The Authors.
Reviewer 2 Report
Comments and Suggestions for Authors
Thank you for your response, the revised version has addressed my concerns. I found a typo, NE5D4L-->NEED4L (line 227). There may be more in the text, please check again before publication.
Author Response
RESPONSE TO REVIEWERS' COMMENTS
Manuscript number: biology-3469525 † Biology (MDPI)
"Targeting Atherosclerosis via the NEDD4L Signaling – A Review of the Current Literature"
REVIEWER #2
General comment
Thank you for your response, the revised version has addressed my concerns.
General Response
Dear Esteemed Reviewer, thank you for taking the time to review our manuscript and for providing invaluable comments and suggestions. We have carefully addressed all your feedback and are grateful that our revised version of the manuscript was significantly improved. We are excited about the acceptance of our manuscript for publication in this critical journal. Your insights and contributions were instrumental in this process, and we genuinely appreciate your attention to detail. Thank you once again for your time and patience in reviewing our article.
Comment #1
I found a typo, NE5D4L-->NEED4L (line 227). There may be more in the text, please check again before publication.
Response
Dear Erudite Reviewer, thank you for this insightful comment and brilliant suggestion. We agree with you, and correcting this typo would undoubtedly enhance our manuscript’s quality and readability. Therefore, we modified NE5D4L into NEDD4L in Lines 227-229 on Page 5 of our revised manuscript document.
Again, thank you for your patience and guidance. Your contributions have been instrumental in reshaping our manuscript. We are proud of the revised version we are presenting to you, and we eagerly anticipate your positive response and approval for its publication.
I, the corresponding author of the manuscript "Targeting Atherosclerosis via the NEDD4L Signaling – A Review of the Current Literature" under the assigned ID biology-3469525, on behalf of my coauthors, once again extend my heartfelt gratitude to the knowledgeable Editor-in-Chief and reviewers for their time and expertise in revising our manuscript. After we addressed their constructive and refined feedback and suggestions, a significantly improved manuscript version emerged. Undoubtedly, their insightful suggestions and feedback have significantly enhanced the quality of our manuscript. We respectfully are at the disposal of the Editor-in-Chief and the Reviewer to address any additional suggestions regarding our publication. Suppose you are satisfied with our newly refined and significantly improved version. In that case, we are eager to anticipate the acceptance of our article for publication in this most critical journal, Biology. Thank you once again for your time and expertise.